

# Non-target and suspect characterisation of organic contaminants in ambient air, Part I: Combining a novel sample clean-up method with comprehensive two-dimensional gas chromatography

Laura Röhler[1,2], Pernilla Bohlin-Nizzetto[2], Pawel Rostkowski[2], Roland Kallenborn[1,3] and Martin Schlabach[2]

[1]Faculty of Chemistry, Biotechnology and Food Sciences (KBM), Norwegian University of Life Sciences, Ås, Norway
[2]Department of Environmental Chemistry, NILU – Norwegian Institute for Air Research, Kjeller, Norway
[3]Arctic Technology Department (AT), University Centre in Svalbard (UNIS), Longyearbyen, Svalbard, Norway

*Correspondence to*: Laura Röhler (laura.rohler@nmbu.no)

**Abstract.** Long-term monitoring of regulated organic chemicals, such as legacy persistent organic pollutants (POPs) and polycyclic aromatic hydrocarbons (PAHs) in ambient air provides valuable information about the compounds' environmental fate as well as temporal and spatial trends. This is the foundation to evaluate the effectiveness of national and international regulations of priority pollutants. Extracts of high-volume air samples, collected on glass fibre filters (GFF for particle phase) and polyurethane foam plugs (PUF for gaseous phase), for targeted analyses of legacy POPs are commonly cleaned by treatment with concentrated sulfuric acid, resulting in extracts clean from most interfering compounds and matrices, and suitable for multi quantitative trace analysis. Such standardised methods, however, severely restrict the number of analytes for quantification and are not applicable when targeting new and emerging compounds as some may be less stable to acid treatment. Recently developed suspect and non-target screening analytical strategies (SUS and NTS, respectively) are shown to be effective evaluation tools aiming at identifying a high number of compounds of emerging concern. These strategies, combining high sophisticated analytical technology with extensive data interpretation and statistics, are already widely accepted in environmental sciences for investigations of various environmental matrices but its application to air samples is still very limited. In order to apply SUS and NTS for the identification of organic contaminants in air samples, an adapted and more wide-scope sample clean-up method is needed, compared to the traditional method which is using concentrated sulphuric acid. Analysis of raw air sample extracts, without clean-up, would generate an extensive contamination of the analytical system with especially PUF matrix-based compounds and, thus, highly interfered mass spectra and detection limits which are unacceptable high for trace analysis in air samples.

In this study, a novel wide-scope sample clean-up method for high-volume air samples has been developed and applied to real high-volume air samples, which facilitates simultaneous target, suspect and non-target analyses The scope and efficiency of the method was quantitatively evaluated with organic compounds, covering a wide range of polarities (logP 2-11), including legacy POPs, brominated flame retardants (BFRs), chlorinated pesticides and currently used pesticides (CUPs). In addition, data reduction and selection strategies for SUS and NTS were developed for comprehensive two-dimensional gas





chromatography separation with low resolution time-of-flight mass spectrometric detection (GC×GC-LRMS) data and applied on real high-volume air samples. Combination of the newly developed clean-up procedure and data treatment strategy enabled the prioritisation of over 600 compounds of interest in the particle-phase (on GFF) and over 850 compounds in the gas-phase (on PUF), out of over 25000 chemical features detected in the raw data set. Of these, 50 individual compounds were identified and confirmed with reference standards, 80 compounds were identified with a probable structure and 774 compounds were assigned to various compound classes. In the here available dataset, 11 hitherto unknown halogenated compounds were detected. These unknown compounds were not yet listed in the available mass spectral libraries.

## 1 Introduction

Air monitoring programmes and case-studies on the environmental fate of anthropogenic pollutants including legacy persistent organic pollutants (POPs) are important tools for environmental risk assessment. Furthermore, data generated in monitoring programmes and case-studies are forming the foundations for integrated modern pollutant regulations as well as the effectivity assessment of international agreements and conventions on POPs (UNECE, 1998; UNEP, 2009a, b; EMEP, 2019). Air measurements of POPs are commonly done using quantitative targeted analytical approaches in combination with highly selective sample clean-up methods often involving destructive sample clean-up with concentrated sulfuric acid ($H_2SO_{4 \ conc.}$), sodium hydroxide (NaOH) or other very selective preparation methods for an effective removal of interfering matrix compounds, originating either from the polyurethane foam (PUF) based sampling material or from naturally occurring air compounds. These methodologies are well-proven and appropriate for most legacy POPs, and therefore recommended as standard methods for POPs in the UNECE-EMEP (United Nations Economic Commission for Europe´s European Monitoring and Evaluation Programme) manual for sampling and chemical analysis (EMEP, 2019). The outcome of the established targeted analytical methods for quantitative measurements of important environmental pollutants are, however, limited as they are covering only a minor part of the currently available list of priority substances identified as potential contaminants (Arnot et al., 2011; Breivik et al., 2012; McLachlan et al., 2014; Vorkamp and Rigét, 2014; Reppas-Chrysovitsinos et al., 2017; NORMAN-network, 2019).

The current demand for various chemicals in technical and day-to-day consumer products is steadily expanding leading to a constantly increasing number of new compounds identified as potential environmental contaminants. In the light of the continuously increasing numbers of chemicals in commerce, the development of single compound quantitative analytical methods for each of these new compound groups is today considered as in-effective, time consuming and expensive. Therefore, there is a strong demand to develop targeted multi-compound analytical methods with the potential supplementation with suspect screening and non-target screening strategies (SUS and NTS). Many of potential emerging contaminants are less persistent and therefore rapidly degraded during destructive sample extraction clean-up and processes (i.e. acid treatment, saponification, lyophilisation, etc.). This limitation is a fundamental restriction for quantitative analyses of such labile compounds as well as identification of hitherto unknown potential contaminants with similar physical-chemical properties.


Hence, there is an obvious incentive for the development of an alternative mild, non-destructive sample clean-up procedure in order to retain the broadest possible range of chemicals and as little as possible interfering matrix in the clean extract. Today, the combination of unspecific sample extraction and clean-up, in combination with high-resolution chromatographic and detection methods is considered a prerequisite for NTS and SUS strategies. In particulate, the application of ultra-high

resolution chromatographic methods (either liquid or gas chromatographic) in combination with  high resolution mass spectrometry (HRMS) enabled the identification and characterization of hitherto unknown environmental contaminants in different matrices (López Zavala and Reynoso-Cuevas, 2015; Alygizakis et al., 2016; Hernández et al., 2015; Masiá et al., 2014; Al-Qaim et al., 2014; Hernández et al., 2007; Rostkowski et al., 2019; Schymanski et al., 2015). Another advanced analytical tool for non-target specific analysis of environmental samples is comprehensive two-dimensional gas

chromatography (GC×GC) coupled to either low-resolution or high-resolution time of flight mass spectrometry (GC×GC-LRMS or GC×GC-HRMS, respectively). Earlier studies have already successfully applied this technology for the identification and characterization of chemical profiles in petroleum product characterisation (Ruiz-Guerrero et al., 2006; Van De Weghe et al., 2006; Arey et al., 2005; van Mispelaar et al., 2005) and in environmental sample analysis (Millow et al., 2015; Ubukata et al., 2015; Mao et al., 2009; Ralston-Hooper et al., 2008; van Leeuwen and de Boer, 2008; Lebedev et al., 2018; Veenaas and

Haglund, 2017). As extracts for SUS/NTS analyses will contain a much broader range of compounds compared with extracts prepared for single compound targeted analyses, it is essential to increase the resolution for both associated chromatographic separation as well as the detection technology compared to traditional target specific quantitative analysis. Comprehensive GC×GC allows the two-dimensional chromatographic separation of analytes from interfering matrix in complex samples (Figure 1). However, also in the GC×GC separation, potential matrix interferences will reduce the quality of the

chromatographic separation. This will also reduce the quality of the collected mass spectra, making the identification of a compound an even more difficult task. Therefore, sample clean-up needs to be optimised for detection and characterization of substances, often present in ultra-trace amounts.

The overall aim of this study was the development of a wide-scope sample clean-up method for high-volume air samples and to develop SUS and NTS strategies, optimised for GC×GC-LRMS data. This novel sample clean-up method was evaluated by

target analytical methods, covering compounds within a wide range of polarities (logP 2-11). The target methods included legacy POPs, brominated flame retardants (BFRs), halogenated agrochemicals, industrial chemicals and currently used pesticides (CUPs). The presented newly developed clean-up method in combination with SUS/NTS strategies was applied on real high-volume atmospheric samples from a background monitoring station in Southern Norway aiming to identify known and new potential chemicals of emerging concern (CECs).





## 2 Experimental Section

### 2.1 Method evaluation samples and real high-volume air samples

The samples of this study were based on (i), the evaluation of the novel wide-scope clean-up method which was based on a recovery test, covering compounds within a wide range of polarities, using spiked surrogate method evaluation samples and

target analysis. And (ii), the application of the novel clean-up method on real high-volume air samples from the Birkenes observatory in combination with the development of SUS and NTS strategies. For both, (i) and (ii), glass fibre filters (GFF; 142 mm in diameter) and PUF plugs (7 cm in diameter, 4 cm in height), commonly used in high volume air sampling (Kallenborn et al., 2013), were used.

For (i), spiked surrogate method evaluation samples (unexposed PUFs and GFFs) were spiked with $^{13}$C labelled standards

representing POPs and CECs analysed within the UNECE-EMEP and AMAP (Arctic Monitoring and Assessment Program) monitoring programmes, as well as native CUPs and pesticide standards, covering a wide range of polarity. A set of three parallel samples of each standard mixture were prepared for quality assurance (POP, Brominated, CUP A, CUP B and CUP C); in total 15 method evaluation samples were prepared (Table 1). A detailed list about all compounds in the used standard mixtures can be found in the SI, Table S2-S5.

For (ii), two dedicated real high-volume air samples were collected during March–April 2015 at an EMEP background monitoring station, the Birkenes Observatory in southern Norway (Aust-Agder 58° 23' N, 8° 15' E, 190 m a.s.l.). The particle phase was collected on GFF (cut-off 10 µm) and the gas-phase on PUF plugs, at a flow rate of ~ 50 m³ h$^{-1}$. The sampling time was 6 days, resulting in sample volumes of 6100 m³ and 6200 m³ respectively. Details on the GFF/PUF high-volume air sampling methodology can be found in Kallenborn et al. (2013).

### 20   2.2 Extraction and sample clean-up

**Extraction.** The spiked surrogate method evaluation samples (i), GFF and PUF combined, were Soxhlet extracted for 8 h in acetone/$n$-hexane (1:1, v/v), resulting in one combined extract for GFF and PUF per sample. The extracts were reduced to 0.5 mL with a Zymark TurboVap evaporator and solvent changed to isooctane before clean-up.

The exposed real high-volume air samples (GFF and PUF) from Birkenes (ii) were spiked with internal standard (ISTD)

mixture (see SI Table S6 for details) and GFFs and PUFs were Soxhlet-extracted separately for 8 h in acetone/$n$-hexane (1:1, v/v), resulting in separate extracts for PUFs and for GFFs for each sample, respectively. After extraction, the individual extracts were reduced to 0.5 mL and the solvent was changed to isooctane. The same steps were carried out for (i) and (ii) with sample blanks (PUFs and GFFs without exposure to outdoor air) for quality assurance (see sect. 2.6).

**Sample clean-up.** For each extract from (i) spiked surrogate method evaluation samples and (ii) real high-volume air samples

from Birkenes as well as sample blanks of (i) and (ii), a custom made three-layer liquid chromatography column was applied for clean-up. The columns consisted of a glass column (l= 250 mm, i.d.= 20 mm), packed with cotton. The bottom layer consisted of a mixture of Z-Sep$^+$ and DSC-18 (2 g each), the middle layer of Florisil (10 g) and the top layer of sodium sulphate





(1 cm). After conditioning the column with an excessive amount of acetone (1.5 x volume of the column), the column was dried, using a vacuum pump (the columns outlet was connected to a vacuum pump). The individual extracts were applied to the dry column and eluted with 80 mL acetonitrile (ACN)/0.5 % citric acid (w/w). After clean-up, the individual extracts were reduced to 0.5 mL with a TurboVap and further concentrated to approximately 200 µL under a gentle stream of nitrogen gas.

After clean-up and prior to analysis, the recovery standard (1,2,3,4-tetrachloronaphthalene, TCN) was added. Details on the used chemicals and equipment can be found in the SI, Table S1.

### 2.3 Target GC-HRMS analysis for method evaluation samples

The samples from part (i) were quantitatively evaluated by target analysis, using GC-HRMS. The detailed quantitative analytical methods applied here are described in Halse et al. (2011) and Kallenborn et al. (2013).

### 2.4 SUS and NTS of real high-volume air samples

The real high-volume air samples (ii) were analysed on a comprehensive high-resolution two-dimensional gas chromatograph coupled to a low-resolution time-of-flight mass spectrometer with unit mass resolution (GC×GC-LRMS, Pegasus® 4D, LECO, St. Joseph, MI, USA system). The GC was equipped with a Restek (Bellefonte, PA, USA) Siltek Guard column (4 m, 0.25 mm), a SGE (Trajan Scientific and Medical, Ringwood, VIC, Australia) BPX-50 (25 m, 0.25 mm x 0.25 µm) first dimension

column and an Agilent J&W (Folsom, CA, USA) VF-1ms (1.5 m, 0.15 mm x 0.15 µm) second dimension column. This samples were processed with the here developed SUS and NTS strategies, optimised and developed for GC×GC-LRMS (sect. 2.5 and 3.2)

Further details on chromatographic specifications are given in the SI.

### 2.5 Data processing/Post-acquisition data treatment

LECOs® ChromaTOF® (V 4.50.8) software, including its advanced features Scripts and Statistical Compare, which also controls the GC×GC-LRMS system, was used for data analysing and processing; including automatic peak finding, spectral deconvolution for coeluting peaks, modulation slice combination and mass spectral searching compared to the used mass spectral libraries. In this study, an in-house custom library with mass spectra of reference standards and $^{13}C/^{2}H$ labelled ISTDs was used in combination with the National Institute of Standards and Technology (NIST) 2014 mass spectral library and the

Scientific Working Group for the Analysis of Seized Drugs (SWGdrug, Oulton (2019)) mass spectral library. For more efficient suspect screening and flagging of potential suspects during data processing, a customised library, containing selected suspect spectra from NIST14, was created. More details about the chosen suspect lists, creation of the customised library as well as the alignment with the final peak list can be found in sect. 3.4 and in the SI.

An in-house developed post-acquisition workflow for GC×GC-LRMS data of the real high-volume air samples was used for

the combined chemical work, target, SUS and NTS (Figure 2 and Figure 3). The level classification concept, developed from Schymanski et al. (2015), describing the levels of classification and identification confidence is currently a gold standard used



for reporting results from SUS and NTS data evaluation. However, the scheme was developed with the LC-HRMS data in mind, and is therefore not directly applicable to the data produced with a GC-MS based methods (Rostkowski et al., 2019). The combination of columns applied in this study (medium-polar combined with non-polar) had an improved matrix separation from compounds of interest, compared to the most common combination (non-polar combined with medium-polar). However,

as stated by Röhler et al. (2020), this column combination is not suitable to use with any available retention indices for further identification confidence. The most comprehensive databases are available for non-polar (5 % phenyl) columns, whereas this study was using a medium-polar column. Limited concepts for retention indices are available for GC×GC, e.g. Mazur et al. (2018) or Veenaas and Haglund (2018), using a non-polar column as first dimension and medium-polar column as second column for GC×GC separation. A new model would be necessary to enable the possibility of retention indices for the column

combination used in this study.

## 2.6 Quality control

Laboratory blank samples were included for both sample types (i) and (ii). The blanks consisted of unexposed PUFs and GFFs and were treated as their respective sample type (i) or (ii) regarding extraction, clean-up and analyses. To ascertain that a detected/reported compound has its origin in the sample (i) or (ii), and not occur in the respective laboratory blank samples for

(i) or (ii), a compound need to exceed an sample concentration factor $\geq 10$ compared to a blank sample in target analysis for (i) or an area factor $\geq 100$ compared to a blank sample in SUS/NTS for (ii).

There were no targeted compounds detected in blanks for part (i). ISTDs, used in SUS/NTS of real high-volume air samples (part (ii)) were used for quality assurance and sample normalisation and not for target quantification. Visual comparisons of peak intensity and intensity ratios from ISTDs were used to identify potential contamination/performance issues of the

GC×GC-LRMS system. This was done for samples and blank samples from (ii) as well as ISTD mixture analysis, which were analysed in between blank samples and samples from (ii).

## 3 Results and Discussion

### 3.1 Evaluation of the novel sample clean-up method

The application of the novel wide-scope sample clean-up method, with a custom three-layer liquid chromatography method,

was quantitatively evaluated with targeted analyses on GC-HRMS of triplicates of unexposed samples (PUFs and GFFs) spiked with a mixture of various compound classes covering a wide range of polarity (logP 2-11). The results show that the novel clean-up method provided extracts of similar cleanness and comparable recoveries for acid-stable POPs as routine methods in monitoring programs for POPs. The recoveries of most of the targeted compounds were over 50 % using the novel clean-up method (Table 2) which is in accordance with the standard QC requirements for this type of analysis. For acid-labile

compounds such as dieldrin, endrin, aldrin, isodrin, heptachlor-*exo*-epoxide, endosulfan I/II/sulphate, ATE (allyl 2,4,6-tribromophenyl ether) and BATE (2-bromoallyl 2,4,6-tribromophenyl ether) the recoveries with the novel clean-up method





were 62-117 % while they are not detected or detected with low recoveries using routine clean-up methods. This shows the advantage of this method to also allow quantitative extraction of acid-labile organic contaminants. More details on recovery of single compounds and relative standard deviations (RSDs) can be found in Table S2-S5 in the SI.

A few of the targeted/spiked compounds had no recorded recovery (i.e. chlorfenvinphos, chlorobenzilate, dichlorvos, endrine aldehyde and etridiazole) or very low recovery (i.e. bromacil and chloroneb). The most probable reason seems to be insufficient elution with the used solvent (ACN/0.5 % citric acid) due to strong/irreversible interactions with Florisil and/or strong Lewis acid/base interactions with Z-Sep$^+$ (Zirconium oxide and C18 coated silica particles).

## 3.2 SUS and NTS identification approach

For compound characterisation, an already reported level classification system for identification confidence by Schymanski et al. (2015) was adopted and optimised for the here used GC×GC-LRMS technique (Figure 2). This level classification is a useful tool to report results from SUS and NTS. The original version was developed for classification of SUS and NTS results from interpretation of LC-HRMS data. This classification strategy provides a suitable platform for a compounds level of identification confidence. The defined confidence levels by Schymanski et al. (2015) are covering identification criteria from accurate mass identification of a compound (Level 5, L5) to direct match with a reference standard (Level 1, L1). As proposed by Rostkowski et al. (2019), the original version of Schymanski et al. is not directly applicable for GC-HRMS, mainly due to different data filtration strategies compared to LC-HRMS. Additionally, in contrast to other previously reported SUS/NTS studies, our work is based on LRMS data, and thus cannot provide accurate masses of compounds of interest. As potential molecular formula and further structural information are not easily available with the here used GC×GC-LRMS technique, we were forced to slightly adjust this level classification scheme for better complying with the needs and limitations of LRMS data treatment. However, our adjusted approach is kept it as close as possible to the original version from Schymanski et al. (2015). An additional Level 0 (L0) was included allowing to distinguish between compounds identified by external reference standards after the original sample analysis (L1) and those compounds identified by ISTDs (L0), added to the sample before sample extraction. Here, direct target quantification of L0 compounds is possible although not further examined in the here reported study. For Level 2 (L2) compounds, a probable structure derived from good library match in combination with a plausible position on the GC×GC 3D surface or an isomer of an available reference standard could be assigned. An example of a L2 compound could be a penta-chlorinated PCB. The mass spectral information is matching well with a penta-chlorinated PCB, however, as there are several possible different penta-chlorinated PCB congeners (n= 47), the individual penta-chlorinated PCB congener could not be identified. For compounds classified as Level 3 (L3), a certain substructure or compound class could be assigned. Here the structure of a compound is not totally clear, but a certain base structure confirmation is possible due to the available information. An example of a L3 compound could be a tentative polycyclic aromatic hydrocarbon (PAH) where the fragment pattern of the mass spectra (MS) was assigned to be a PAH with a possible molecular formula. Since there are too many possible PAHs (n > 100) with various structures matching the given MS and molecular formula, it is only possible to assign a compound class to this compound. Compounds classified as Level 4 (L4) are



only defined by a possible molecular formula or by characteristic halogen cluster/-s. They do not match any MS in the used MS libraries. All peaks, which were matching the criteria for SUS and/or NTS during DP (Figure 3 before reaching A) were classified as Level 5 (L5), mass spectra of interest.

In comparison to target analysis, developed for the highest confidence level of identification, SUS and NTS results have
different confidence levels as described above. In target analysis, isotope dilution analysis with ISTDs is, beside others, a commonly applied technique (EFSA, 2010; European_Commission, 2017). The hereby used specific sample clean-up for those selected compounds removes the bulk of disturbing matrix and other potential deteriorating issues with potential effects on the chromatographic separation,  Hence, the results are reported as validated concentration levels in table form for all targets analytes  (Figure 2, Level 0). Whereas, for SUS and NTS a more general sample clean-up procedure is necessary which often
does not remove all interfering matrix. These SUS/NTS results are identified as extensive lists of  relevant peaks (often ≥ 20000 peaks), typically detected via retention time (RT) and full scan mass spectra information (Rostkowski et al., 2019). Usually, the original peak list identified automatically by the analytical software, need to be systematically reduced and categorised  according to the above described confidence identification criteria (Figure 2, Level 1-5). Such a data reduction is necessary for a sound interpretation of the results (Figure 3). As described in section 2.5 the instrumental software generates
an initial peak list containing 10000s entries. In order to have an efficient data treatment, it is required to priorities properly and reduce the originally long peak lists. This first reduction step is to identify and remove compound signals, which are also occurring in sample blanks. Based on the available software tools a data processing workflow was applied including compound identification with MS libraries, identification of compounds which occur in one or more samples, identification of halogen isotopic clusters or other specific ions (e.g. m/z 149 as base peak for phthalates, etc.). After these automated processes, the
received peak list was further reduced by manual or semi-automatic inspections resulting in a shortened peak list, corresponding to previously defined quality thresholds. To increase the level of identification confidence, manual inspection of each peak is necessary. This evaluation step is very time consuming and thus limit the number of compounds for which such semi-automated/manual inspection could be performed.

### 3.2.1 Automatic blank filtration

The first step in reducing the originally long peak lists produced by deconvolution of raw data is to identify and remove compound signals which are also occurring in sample blanks. Since SUS and NTS at this stage is resulting in qualitative/semiquantitative rather than quantitative results, the exact compound concentration in the collected air samples and blanks is unknown. Therefore, blank compound filtration is based on comparison of signal areas only. In order to compensate for response variation occurring between real sample extracts and method blanks, a high threshold for detection is applied,
considerably higher as utilised for traditional target analysis. In our case, a compound in a real sample must exceed an area factor ≥ 100 compared to a blank sample to be confirmed as a detected compound.

After automatic sample blank filtration for NTS/SUS analysis, the peak list of the air samples from Birkenes still covered a large number of compounds also confirmed in sample blanks. This poor efficiency of automated blank filtration can be





explained by the differences in peak distribution profiles for the different blank samples and for the average of the blank samples compared to the real samples. Only 50-75 % of the identified blank contaminants were identical in the different blank samples. However, the automatic filtration procedure reduced approximately 10 % of the total peak number (reduction from about 26000 to 24000 peaks for PUF sample and 25000 to 22000 peaks for GFFs). Further strategies for peak filtration had to

be applied to reduce the number of peaks. Such an effective filtration is necessary providing a suitable platform for priority compound identification (Figure 3, to reach A) and classification of the different confidence levels (Figure 2, L1-L5).

During initial data processing, the here used ChromaTOF® software is automatically finding all relevant signals/peaks, deconvolute coeluting mass spectra, combining modulation slices and comparing this spectral information against the set of chosen MS libraries. Hereby, it may happen that one signal in the chromatogram is associated with several peak markers, e.g.

if the peak width is broader than the used specifications for automatic peak finding or peaks are tailing. Unfortunately, the automated deconvolution algorithm from ChromaTOF can mark a single compound with several peak markers, which was shown in a study by Lu et al. (2008). Due to these limitations, the total number of originally detected compounds is usually lower than the number of peak markers. First during comprehensive manual inspection (Figure 3, A) these additional false peak markers will be discovered and peak lists corrected for duplicate peak markers.

**3.2.2 SUS data processing workflow**

In this study, the data processing strategy (DP) was split in two parts, SUS (Figure 3, I) and NTS (Figure 3, II). After the initial automated peak identification, the peak lists from both DP approaches were merged to one L5 list for manual check on identity (Figure 3, A) and further level of identity confidence classification.

During SUS DP (Figure 3, I), all MS of the automatically detected peaks were searched against the MS libraries reference

information for SUS (in-house custom libraries of reference standards and ISTDs, customised suspect library as described in sect. 2.5 and SWGdrug Oulton (2019) mass spectral library). Added ISTDs were identified (L0), as well as sample blank compounds. A second blank filtration was performed and only compounds which are exceeding an area of factor $\geq 100$ compared to the sample blank were kept for further inspection. As described in the previous section 3.2.1, this high threshold is necessary to compensate for different sample volumes and unknown variation of response between extracts. After blank

filtration, all peaks with a forward match of $\geq 70$ % to the MS listed in custom suspect libraries for SUS were identified (Figure 3, I: preliminary L5 list). These peaks from "I: preliminary L5 list" (Figure 3) were further processed by including the entire NIST14 MS library in addition to the previously applied custom suspect libraries, to ensure the quality of the library identification procedure (Figure 3, I: L5 list to check manually on identity). Applying this procedure, approx. 600 suspects were identified in the PUF and approx. 400 suspects in the GFF samples. These signals were only identified by MS library

matching, without manual check of their identity, the confidence level of identification is here L5 and for found ISTDs and their respective native compound, L0 (Figure 2). In order to improve the confidence level of identification for these compounds, the manual check on right identification is required as the next step (Figure 3, A; in combination with results from NTS).

### 3.2.3 NTS data processing workflow

For NTS DP (Figure 3, II), LECOs statistical compare® tool for the identification of all compounds occurring in both PUF or both GFF samples was applied. With this approach, it was possible to reduce the peak lists from approx. 30000 to 3800 peaks for PUF and from approx. 25000 to 5000 peaks for GFF samples. After the initial automatic blank filtration (see sect. 3.2.1),

DP with the NIST14 and suspect libraries as well as applying NT scripts for the identification of specific compounds of interest (i.e. halogenated etc.) was performed. The resulting peak list was further reduced to approx. 1000 peaks per sample. These NT scripts, written in Visual Basic, were applied during DP to identify brominated and chlorinated compounds based on their isotopic clusters, as well as PAHs, phthalates and nitro compounds with the help of recognizable features in fragmentation patterns (Hilton et al., 2010). These scripts are especially useful to detect compounds which would be overlooked by low MS

library match or not listed in the used MS libraries. In addition, a second blank filtration were performed and only compounds which are exceeding an area of factor $\geq 100$ compared to the sample blank were kept for further inspection. Like in SUS DP, also during NTS DP it was necessary to reduce the number of peaks for manual inspection. As a final method, all peaks identified with NT scripts and all peaks with a forward match of $\geq 80$ % to the MS libraries (NIST14 and suspect libraries) were kept for further processing. Hereby it was possible to identify approx. 550 compounds in the PUF sample and approx.

400 compounds in the GFF sample with NTS DP. Those identified compounds were classified as L5 and ISTDs and their respective native compound, L0 (Figure 3, II: L5 list to check manual on right identity).

Similar to SUS, manual check on the right identity of these NTS L5 compounds is needed in order to increase the level of identification confidence since all confirmations are only based on MS library comparisons or NT script filtrations. For manual inspection of each compound and further level classification, the lists from SUS and NTS were merged to one list for a more

effective proceeding (Figure 3, A).

Both DPs, SUS and NTS, were using the forward match percentage to MS library entries to reduce the number of peaks which require manual inspection. In this step, the quality of a MS from a compound is of high importance to match a MS library entry and thus be kept for further processing. The quality of a MS of a compound is not only affected by interferences or S/N ratios, the quality might also be affected from the unit mass resolution of the used GC×GC-LRMS instrument. In particular, the

limited unit mass resolution of the used GC×GC-LRMS has negative consequences for MS of compounds with higher mass defects, e.g. brominated, higher chlorinated or mixed halogenated compounds. Even when acquired under optimal conditions, the obtained MS are not identical to reference MS from the NIST14 MS library (Figure 4) and, hence, those compounds would be rejected during DP, due to low match percentage to NIST14 library. The used NT scripts used during DP, developed by Hilton et al. (2010), were specifically developed for MS obtained by LECOs GC×GC-LRMS for the identification of isotopic

clusters of brominated and chlorinated compounds and were used as an tool during DP for the identification of compounds of interest for manual inspection.

In addition to the MS quality affected by the unit mass resolution of the ToF-MS detector, lower library match could also be caused by different fragmentation patterns compared to MS from the NIST14 library, which were obtained with quadrupole



mass filter in electron ionisation mode. Also here it was possible that compounds of interest could be rejected during a DP step due to low match percentage to a NIST14 MS.

Further factors may limit the positive identification of a compound including potential loss during sample clean-up. Our sample clean-up method was optimised for the analysis of compounds covering a wide range of polarity for GC×GC-LRMS analysis.

However, the substantial loss of substances purely adsorbing and accumulating on PUF/GFF sampling materials cannot be excluded. Furthermore, compounds may degrade, evaporate or not elute from used adsorbents during sample clean-up. During GC×GC-LRMS analyses, thermolabile substance may degrade in the injector or irreversibly bound/degraded on the chromatographic column. Furthermore, compound specific low sensitivity in the here used positive electron ionisation mode may prevent the positive identification of a possible target compound.

In the here chosen DP strategy, all confirmed compounds need to match all used selection criteria. However, the priority criteria need individual fine tuning for each data set examined for avoiding false positive and false negative listings as well as minimize the occurrence of blank compounds. However, even after following this comprehensive data processing protocol, it cannot be excluded that unconfirmed or excluded substances does not occur in air from Birkenes, southern Norway.

### 3.3 Number of detected and classified compounds

After comprehensive peak filtration from raw data to a reduced peak list for manual inspection, all remaining compounds were initially classified as L5 (mass spectra of interest) (Figure 3: A) and, respectively all compounds, identified with ISTDs as L0. The compounds classified as L5 are further checked manual on their identity to reach a higher level of identification confidence. For some compounds, with high match percentage compared with the reference MS libraries and recognisable m/z pattern/-s in the MS, this check on right identification is a straight forward procedure for classification as L2 or L3 compounds.

Others, with less characteristic m/z patterns, or just an identification due to their inherent halogen isotopic pattern, might be classified as L3 or L4 (Figure 2). The procedure for the correct classification of such substances is time consuming and requires comprehensive scientific experience. Before comparing compounds to in-house and/or new reference standards, L2 and L3 compounds were, in addition to the automatic blank filtration during initial data processing, manually checked against sample blanks and ensured that these compounds have an area, which exceeds the area threshold (factor ≥ 100). This manual blank

check is essential, since the automatic blank filtration routine during DP may lead to missing compounds (low match factors between the blank and the real sample), partly caused of coelution or matrix related retention time shifts. After this initial step, further characterisation of potential compounds based on sales numbers, inherent physical chemical properties (adsorption, transformation, reactivity), application sources and profiles, seasonal patterns etc, may be beneficial in addition to confidence level determination (L0, L1, L2, L3, L4 or L5).

For the here studied high-volume air samples from the Birkenes observatory, the merged L5 list from SUS and NTS available for manual inspection (Figure 3, A) contain almost 1500 compound suggestions: over 600 compounds from the GFF extracts (particulate phase), and over 850 compounds from the PUF extracts (gaseous phase). More than 50 % of these compounds could be further identified and classified to L4, L3 and L2 during manual inspection of MS. This was possible for 350





compounds from the GFF and for 655 compounds from the PUF. All L2 and L3 compounds were manually checked against the blank sample before comparison to new and in-house reference standards. For quality assurance, all reference standards were analysed with the same GC×GC-LRMS method as the air samples , as well as analysing a reference mixture of ISTDs to account for retention time shifts (Figure 3, B). Hereby, five compounds were confirmed with ISTDs to L0 (1/4 GFF/PUF) and

45 compounds with reference standards to L1 (12/33 GFF/PUF). In addition, 80 compounds were classified as L2 (21/59 GFF/PUF) and 774 compounds as L3 (290/484 GFF/PUF). The remaining 81 compounds were characterised as L4 (17/64 GFF/PUF) compounds as summarised in Figure 3, C and Table 3.

The L2 compounds include 11 potential PCBs. For those compounds the exact number of congeners might deviate since single reference standards for each PCBs congener were not analysed. Polycyclic aromatic compounds (PAC) was the largest sub-

group of L3 compounds (see Figure 6). Unknown halogenated compounds, which did not have any MS library match, were included in L4. An overview about the distribution of L0–L4 compounds in the GFF and PUF can be found in Table 3. The complete peak list of L0–L4 compounds is available in the Excel-SI spreadsheet.

From 45 compounds, classified as L1, 22 compounds are listed in one or more suspect lists, and from 80 compounds, classified as L2, resemble 28 compounds similarity to one or more suspect lists (Table 3). As L2 compounds are not confirmed with

reference standards, matches to suspect lists are slightly uncertain and compounds listed as L2 in Excel-SI may also represent different isomers.

The here chosen priority suspect lists were selected for the identification of long-range atmospheric transport potential (LRATP) of CECs and hitherto unidentified CECs. However, the chosen suspects do cover the bulk of legacy POPs, CECs previously analysed at the Birkenes observatory and a large number of CUPs and non-regulated chemicals, especially own

measured MS in the customised self-build libraries. The chosen suspects list are considered as relevant for Arctic air samples and suspect prioritisation lists originate from different authors (Reppas-Chrysovitsinos et al., 2017; Brown and Wania, 2008; Coscollà et al., 2011; Hoferkamp et al., 2010; Howard and Muir, 2010; NORMAN-network, 2019; Vorkamp and Rigét, 2014; Zhong et al., 2012) as well as self-build in-house suspect libraries (Table 3). A short summary about data alignment of used suspect lists and findings in our samples can be found in the SI.

The compounds and compound groups identified in the air samples from the Birkenes observatory in this study are grouped in three groups: (i) legacy POPs and PAHs, (ii) known CECs and (iii) new potential CECs not previously reported in southern Norway/Birkenes (status October 2019). In addition to 36 already reported organic contaminants at Birkenes (incl. legacy POPs and known CECs), 92 new potential CECs with match to reference standards (L1) or probable structures (L2) were identified (64 in PUF and 28 in GFF samples). It is interesting to note that 11 chemicals were common to the GFF and PUF

sample. 29 of the new potential CECs have a LRATP according to the Stockholm convention (UNEP, 2009a), half-live in air exceeding 2 days, and may, hence, undergo long-range atmospheric transport.

Overall, 39 compounds, identified as L0, L1 or L2, were also detected in high volume air samples from the Zeppelin station (Ny-Ålesund) in Svalbard, using the same analytical approach as in this study (Röhler et al., 2020).





A complete overview can be found in the Excel-SI spreadsheet, including information on the complementary findings in Arctic air samples, physical-chemical properties, additional information from literature search as well as further parameters on environmental properties (incl. persistence, bioaccumulation and toxicity (PBT) classification by REACH (European Parliament, 2018) and Stockholm convention (UNEP, 2009a), Table S7).

## 3.4 Identified compound groups

As summarised in Figure 5, identified compounds were grouped in different compound classes and arranged as previously detected or previously not detected in air samples at the Birkenes observatory (only including L0, L1 and L2 compounds). For approximately 2/3 of the identified compounds, an application purpose could be identified and are discussed in detail in the following sections.

### 3.4.1 Legacy POPs and PAHs in air from Birkenes

In total, 23 legacy POPs and PAHs were identified as L0, L1 or L2. The L0 and L1 were hexachlorocyclohexanes (α-HCH and γ-HCH), HCB, *p,p'*-DDE, *p,p'*-DDT, PCB 153, dieldrin, *trans*-nonachlor and a metabolite of heptachlor (heptachloro *exo*-epoxide) and three PAHs, routinely measured at Birkenes, such as biphenyl, fluorene and benzo[*ghi*]fluoranthene (UNEP, 2009a). An extensive list of PAHs was detected showing their presence in air samples from Birkenes, but only a few single PAH reference standards were available for analyses and hamper the identification of individual PAHs. Most of the detected PAHs were therefore classified as L3 (section 3.4.4). In addition, 11 PCB congeners were classified as L2. Besides dieldrin and heptachloro *exo*-epoxide, the remaining legacy POPs are regularly measured using target methods in the Norwegian monitoring programme for long-range transported atmospheric contaminants (Nizzetto, 2016) at the same monitoring station. The detection of those compounds with our novel wide-scope sample clean-up method, combined with SUS and NTS characterisation method in real air samples provides additional confidence for the quality of the here reported comprehensive analytical strategy.

### 3.4.2 Known CECs

The presence of four known CECs (L0, L1 and L2), recently reported in Birkenes air samples, where also confirmed by the here applied approach (Nizzetto, 2019). These includes BFRs, pentabromotoluene (PeBT, L2) and hexabromobenzene (HBB, L0) as well as OPFRs, triisobutyl phosphate (TBP, L1) and tris(1,3-dichloro-2-propyl)phosphate (TDCPP, L1). In addition to the monitored OPFRs, it was possible to detect nine isomers of previously monitored OPFRs as L2. Two positional isomers of tris(4-isopropylphenyl) phosphate (TiPPP), three isomers of di(isopropylphenyl)phenyl phosphate, one isomer of isopropylphenyl diphenyl phosphate as well as one positional isomer of tris(2-chloroisopropyl)phosphate (TCPP), one isomer of cresyl-diphenyl phosphate and one TBP related isomer as L2. The six isopropylphenyl phosphate congeners are all part of the technical mixture of TiPPP.



### 3.4.3 New potential CECs

In addition to identification of legacy POPs, PAHs and known CECs in air samples from Birkenes, it was possible to detect 90 new potential CECs that to our knowledge have not been reported previously in air samples from this region. Most of these new potential CECs (n=62), identified with match to reference standards (L1) or probable structure (L2), were detected in the

gas phase (PUF) while 28 were detected in the particle phase (GFF).

**Compounds with LRATP.** According to half-life data ($t_{1/2}$(air)) of the AOPWIN model of US EPAs EPIsuite program (U.S.EPA, 2019), 29 of the detected new potential CECs have a LRATP according to the Stockholm convention criteria (UNEP, 2009a), $t_{1/2}$(air) exceeding 2 days.

Of these 29 compounds, 14 were identified as L1 (4/10 GFF/PUF; of those are 4 common to GFF and PUF) and 15 compounds

were identified as L2 (4/11 GFF/PUF). Structures, sample, name and CAS for L1 compounds can be found in Figure 4, all further information is available in the Excel-SI spreadsheet.

The four L1 compounds, which were identified both in the GFF and PUF samples were benzenesulfonamide (BSA), p-toluenesulfonamide (pTSA), 2-methyl-9,10-anthraquinone (2-MAQ) and 4H-cyclopenta[def]phenanthren-4-one. BSA and pTSA have similar molecular structures, since BSA is the parent compound of pTSA. BSA is used as an industrial intermediate

in the synthesis of widespread products like disinfectants, dyes or photochemical products and pTSA is used as a fungicide in paints and coatings or as a plasticiser (ECHA, 2019b; Naccarato et al., 2014; Herrero et al., 2014). Since BSA and pTSA could be used in many widespread products, a local source cannot be excluded. The identified 2-MAQ is a potential wood combustion product, an intermediate in industrial production of coating products, inks, toners, laboratory chemicals and explosives, and used for the production of plastic products (Czech et al., 2018; Lui et al., 2017; Vicente et al., 2016; ECHA, 2019a). It is also

possible that 2-MAQ could be formed through atmospheric reactions (Alam et al., 2014). All three oxy-PAHs, 2-MAQ and 4H-cyclopenta[def]phenanthren-4-one (identified in GFF and PUF) and 9,10-anthraquinone (PUF only), are related to emissions of diesel and petrol vehicles (Karavalakis et al., 2010; Alam et al., 2014, 2013). 4H-Cyclopenta[def]phenanthren-4-one and 9,10-anthraquinone are also identified as oxidation products of PAHs (Singh et al., 2017). The three identified oxy-PAHs are known air contaminants, but to our knowledge never been measured in Norwegian background air samples before.

To understand the origin of these oxy-PAHs, further research is necessary, e.g. diagnostic ratios to distinguish between different sources (Alam et al., 2013).

The remaining five L1 compounds (only detected in PUF) were two intermediates, 1,4-benzenedicarbonitrile (terephthalonitrile) and 1-methyl-2-nitrobenzene (2-nitrotoluene), the biodegradation product tetrachloroveratrole as well as two combustion products, 1-methoxy-2-nitrobenzene (2-nitroanisole) and 2-naphthalenecarbonitrile. Terephthalonitrile might

be an intermediate for the production of the pesticide dacthal (Meng, 2012) and was detected together with two isomers of terephthalonitrile (probably positional isomers), which were classified as L2. 2-Nitrotoluene is used as an intermediate for the production of azo dyes and other dyes, rubber chemicals, agriculture chemicals, pharmaceuticals and explosives (IARC, 2013; ECHA, 2008). The presence of 2-nitrotoluene may also be a degradation product of explosives like TNT (trinitrotoluene)



(Mohsen et al., 2013). A possible local source could be a shooting range (6 km south-westerly) or military training areas, which is approximately 30 km south-westerly from the Birkenes observatory (NOU, 2004). The pesticide metabolite, or bacterial biodegradation product tetrachloroveratrole is formed during bleaching of wood pulp or chlorination of wastewaters in the pulp and paper industry (GovCanada, 2019; Su et al., 2008; Arinaitwe et al., 2016). Tetrachloroveratrole is a known priority

pollutant, found and monitored even in the Arctic (Su et al., 2008), but previously not reported in southern Norway background air. 2-Nitroanisole is mainly derived from combustion processes but can also be formed by atmospheric reactions (Stiborova, 2002). Large quantities of 2-Nitroanisole were released into the atmosphere in the course of an accident at the Hoechst plant, Germany in 1993 (Weyer et al., 2014). 2-Naphthalenecarbonitrile is related to plastic combustion, e.g. ABS (acrylonitrile-butadiene-styrene) plastic or polyester fabrics (Moltó et al., 2009; Watanabe et al., 2007; Wang et al., 2007; Moltó et al., 2006)

but can also be used for the bluing of steel surfaces (Stefanye, 1972). The corresponding isomer 1-naphtalenecarbonitrile was classified as L2. Other compounds identified as L2 can be found in the Excel-SI spreadsheet.

**Compounds without LRATP.** The other group of new potential CECs detected in this study (n=61) do not have LRATP, according to the Stockholm convention criteria (UNEP, 2009a), $t_{1/2}$(air) need to exceed 2 days. The origin of these compounds is still considered to be through LRAT as Birkenes is a background monitoring station where background air is being measured.

The presence of these compounds at Birkenes is therefore itself an evidence for LRAT of these compounds. It shows a limitation of modelling calculations for LRATP. The results of this study can be compared with data from the Zeppelin observatory on Svalbard (Arctic background air samples) reported earlier (Röhler et al., 2020). In brief, 16 of 17 L1 compounds without LRATP (all compounds in Table 5, except 3,6-Dimethylphenanthrene) from the Birkenes dataset were also confirmed in the Arctic air samples, further confirming LRATP of these compounds. For more details see Excel-SI.

Overall, 61 new potential CECs without LRATP were classified in Birkenes air samples, 17 compounds were identified as L1 (5/12 GFF/PUF; 4 are common to GFF and PUF) and 44 compounds classified as L2 (15/29 GFF/PUF; 3 are common to GFF and PUF). For L1 compounds, CAS, name, sample and structure are listed in Table 5, and further information on all compounds identified can be found in SI Excel-SI.

Four oxy-PAHs, 1,2-BAQ, BPone, BAone, 9-Fone, and one PAH, 3,6-DMPH, have previously been detected in particle related

samples from three southern European cities, with highest concentrations during winter (Alves et al., 2017), but to our knowledge have not been previously measured in south Norwegian air samples. 3,6-DMPH and 9-Fone were found in the PUF, BPone in the GFF and 1,2-BAQ as well as BAone in the GFF and PUF sample. The identified PAH and four oxy-PAHs were all previously detected in wood combustion experiments (Czech et al., 2018) and a local sources cannot be excluded. A further group of compounds, consisting of three terphenyl isomers (*o,m,p*), were previously detected during pyrolysis and

combustion experiments of polyether fabric (Moltó et al., 2006). The commercial mixture of all three terphenyl isomers is used for heat transfer and storage agent in industrial processes. Also applications as textile dye carriers and as intermediate of non-spreading lubricants are reported (Netherlands, 2002). All three terphenyl isomers were identified in the PUF sample and m-terphenyl was in addition to that, also detected in the GFF sample. The terphenyls were to our best knowledge never before analysed in air samples from southern Norway but were part of a larger screening study from Oslo in 2018. In that study,





terphenyls were found in indoor air, sewage water and sediment samples, indicating their widespread emission to the environment (Schlabach, 2019).

Carbazole may mainly be used in carbazole containing polymers (PVK, poly(-N-vinylcarbazole)), which could be used in photovoltaic devices or in semiconducting polymers (Zhao et al., 2017; Grazulevicius et al., 2003). This compound is also

used in the production of various pharmaceuticals (Zawadzka et al., 2015). Carbazole was identified in both GFF and PUF sample. For two identified wood preservatives, dichlofluanid and IPBC, a local source cannot be excluded. IPBC is also used in cosmetics and personal care products (ECHA, 2019c, d). Both compounds were detected in the PUF sample. Triallate, which was detected in PUF sample, is used as agriculture pesticide (herbicide). While never being detected in air samples from southern Norway, there was a previous finding in air samples from Manitoba (Canada) during winter, suggesting relatively

high persistence in air and possibly LRATP (Messing et al., 2014). A major methylation product of 2-mercaptobenzothiazole (2-S-BTH), 2-Me-S-BTH, could be identified in the PUF sample. 2-S-BTH is used as vulcanisation accelerator in rubber of car tires, shoes, cables, rubber gloves and toys (Herrero et al., 2014; Leng and Gries, 2017). Due to its widespread use, the finding of 2-Me-S-BTH could be affected by local sources.

### 3.4.4 Summary for Level 3 compounds

A large number of L3 compounds (tentative candidates; n=774) were identified. After grouping those L3 compounds in classes, the largest groups of compounds are PACs (polyaromatic compounds), carbonic acid esters and phthalates. Other detected esters and a few halogenated compounds were two minor groups. All further compounds were grouped as miscellaneous (Figure 6). The list of L3 compounds can be found in the SI Excel-SI.

### 3.4.5 Summary for Level 4 compounds

In the group of L4 compounds, 81 possible molecular formula and unknown halogenated compounds could be detected. Of these, 11 were classified as potential unknown halogenated compounds (2/9 GFF/PUF) and the other 70 compounds only with possible molecular formula (15/55 GFF/PUF; 2 are common to GFF and PUF). The detected unknown halogenated compounds did not match MS from NIST14 or in-house MS libraries. It was, however, possible to extract a potential content of chlorine and/or bromine, a potential molecular weight and structural fragments from the given LRMS spectra. For further identification,

to receive more structural information or a potential molecular formula, investigation on HRMS instruments is required. The list of detected L4 compounds can be found in SI Excel-SI.

## 4 Conclusions

A comprehensive sample clean-up method is one of the key factors for successful SUS and NTS approaches. An ideal method removes interfering matrix and in the same time keep a maximum number of compounds of interest in the extract. In this study,

a novel sample clean-up method has been developed and tested on spiked samples and real air samples. The results demonstrate

that this method is promising in target as well as SUS and NTS analyses of regulated and emerging organic compounds in air samples. The recoveries for legacy POPs and BFRs were comparable to those obtained with the traditional acid clean-up method, but with the possibility to quantify an extended range of compounds including the acid-labile POPs and BFRs. The GC×GC-LRMS analyses in combination with the newly developed SUS/NTS data evaluation strategies on real air samples resulted in the identification of 90 new potential CECs, here detected in southern Norway for the first time. With the application of ISTD to SUS and NTS, we extended the SUS and NTS approach into potential quantitative target analysis.

In order to increase the effectiveness of future SUS and NTS studies in air, expanding the suspect library with entries of relevant airborne contaminants is considered as essential. GFF and PUF-based high volume air sampling is a widely used air sampling technique, but the polyurethane polymer used in the foams generates a massive load of PUF related matrix (often more than 20 000 compounds) which need to get removed during sample clean-up or during post acquisition data filtration. Reducing this load by developing cleaner PUFs or replacing PUF with another adsorbent is an important next step in further development of SUS/NTS methods for air samples. In future work, the application of GCxGC-HRMS would be an important step for further improvement of the presented SUS/NTS method as it enables structure elucidation of CECs not yet present in MS libraries. In addition, the application of retention indices and retention index prediction data would provide additional information for the selection of the most likely compound structure.

**Competing interests**

The authors declare that they have no conflict of interest.

**Acknowledgment**

Special thanks to Anders Borgen, NILU for his help with target CUP A-C GC-HRMS analysis and quantification.
Compound structures were created using ChemOffice19 (PerkinElmerInformatics, 2019).
LogP and logD values were created using JChem for Excel (ChemAxon, 2019).

**Author contribution**

LR, MS, PBN and RK developed the idea behind this study.
LR performed chemical work, analysis, created the figures and wrote the paper.
MS and PBN provided guidance and contributed to the paper preparation
PR provided guidance for Z-Sep$^+$/C18 method development and paper preparation
RK provided financial support from internal NMBU funding, academic guidance and contributed to the paper preparation
All authors read and approved the submitted manuscript.





**Financial support**

This study was funded by NMBU, Norwegian University of Life Sciences, Ås with an internal PhD grant, NILU, Norwegian Institute for Air Research, Kjeller and the Norwegian Ministry of Climate and Environment through two Strategic Institute Programs, granted by the Norwegian Research Council ("Speciation and quantification of emerging pollutants" and "New measurement methods for emerging organic pollutants").



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





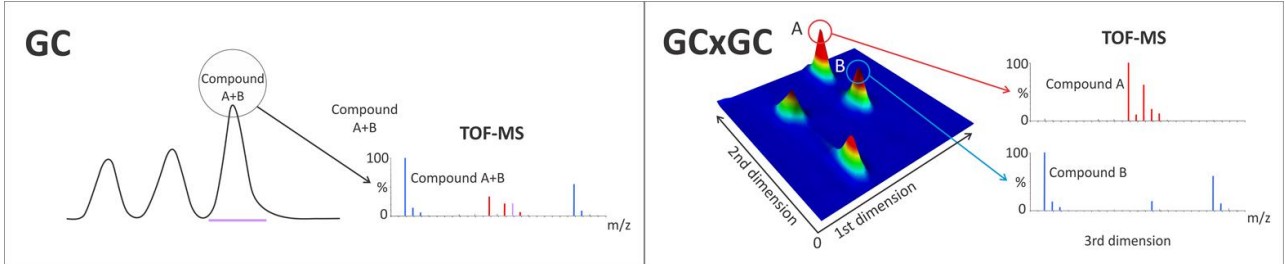

**Figure 1: GC separation compared to comprehensive GC×GC separation (Röhler et al., 2014).**



**Table 1: Spiked standard mixtures for method evaluation samples.**

| Sample type (Set of 3 parallels) | Standard mixture native compounds | Standard mixture [13]C-labelled compounds |
|---|---|---|
| POP | - | POP |
| Brominated | BFR | BFR |
| CUP A | Mix 1 | - |
| CUP B | Mix 2 | - |
| CUP C | Mix 3 | - |





**Table 2: Summary of average recovery rates [%] for legacy POPs, BFRs, CUPs and CECs.**

| Compound class | Average recovery from 3 parallels [%] | Number of compounds |
|---|---|---|
| POPs | 50 – 117 | 40 |
| BFRs | 45 – 92 | 19 |
| CUPs and CECs | <20 | 2 |
| | 20-50 | 11 |
| | >50 | 31 |





**Figure 2: General strategy and levels for identification confidence for GC×GC-LRMS. Adapted from Schymanski et al. (2015).**







**Figure 3: Data processing workflow and peak reduction during level classification.**



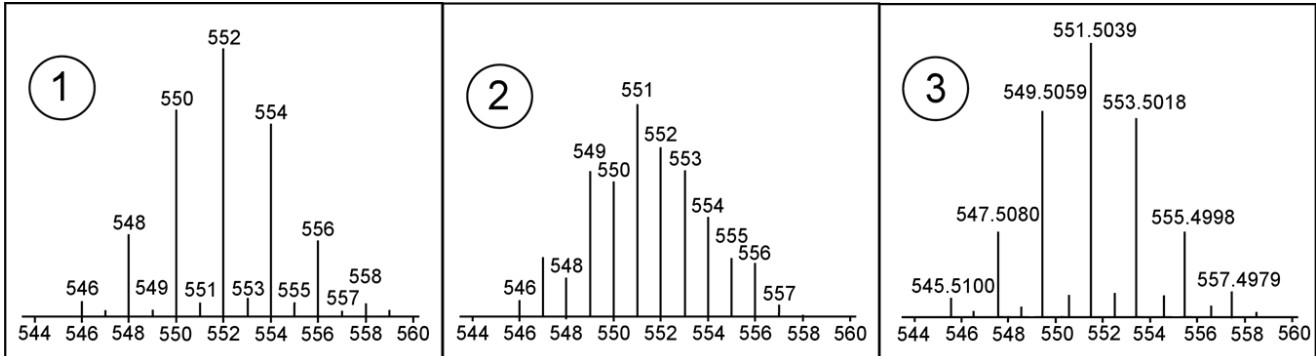

**Figure 4: 1: Isotope cluster of hexabromobenzene (HBB) in NIST14, 2: own measured HBB on GC×GC-LRMS and 3: HRMS isotope cluster HBB.**





**Table 3: Overview of the L0 – L4 compounds, classified in air samples from Birkenes (southern Norway).**

| Level | Compounds classified | PUF samples | GFF samples | Common to PUF and GFF | Found in suspect lists |
|-------|---------------------|-------------|-------------|----------------------|----------------------|
| **L0** | 5 | 4 | 1 | 1 | 4 |
| **L1** | 45 | 33 | 12 | 10 | 22 |
| **L2** | 80 | 59 (11 PCBs) | 21 | 4 | 28[a] |
| **L3** | 774 | 484 | 290 | _[b] | _[b] |
| **L4** | 81 | 64 (9 unknown halogenated) | 17 (2 unknown halogenated) | 2 | _[b] |

a: showing similarity to suspect lists, isomer not confirmed; b: not applicable





**Figure 5: Overview of detected compounds confirmed with reference standards (L0 and L1) and probable structures (L2).**





**Table 4: Structure overview of L1 compounds, classified as new potential CECs with LRATP.**

| Name/ CAS/ Sample | Structure | Name/ CAS/ Sample | Structure |
|---|---|---|---|
| Benzenesulfonamide (BSA)/ 98-10-2<br><br>GFF (particle phase) and PUF | | 1,4-Benzenedicarbonitrile (Terephthalonitrile)/ 623-26-7<br><br>PUF (gas phase) | |
| p-Toluenesulfonamide (pTSA)/ 70-55-3<br><br>GFF (particle phase) and PUF | | 1-Methyl-2-nitrobenzene (2-Nitrotoluene)/ 88-72-2<br><br>PUF (gas phase) | |
| 2-Methyl-9,10-Anthraquinone (2-MAQ)/ 84-54-8<br><br>GFF (particle phase) and PUF | | Tetrachloroveratrole/ 944-61-6<br><br>PUF (gas phase) | |
| 4H-Cyclopenta[def] phenanthren-4-one/ 5737-13-3<br><br>GFF (particle phase) and PUF | | 1-Methoxy-2-nitrobenzene (2-Nitroanisole)/ 91-23-6<br><br>PUF (gas phase) | |
| 9,10-Anthraquinone/ 84-65-1<br><br>PUF (gas phase) | | 2-Naphthalenecarbonitrile/ 613-46-7<br><br>PUF (gas phase) | |





**Table 5: Structure overview of L1 compounds, classified as new potential CECs without LRATP.**

| Name/ CAS/ Sample | Structure | Name/ CAS/ Sample | Structure |
|---|---|---|---|
| 1,2-Benzanthraquinone (1,2-BAQ)/ 2498-66-0 <br><br> GFF and PUF | | 9-Fluorenone (9-Fone)/ 486-25-9 <br><br> PUF (gas phase) | |
| 6H-Benzo[*cd*]pyren-6-one (BPone)/ 3074-00-8 <br><br> GFF (particle phase) | | 3,6-Dimethylphenanthrene (3,6-DMPH)/ 1576-67-6 <br><br> PUF (gas phase) | |
| 1,9-Benz-10-anthrone (BAone)/ 82-05-3 <br><br> GFF and PUF | | Dichlofluanid/ 1085-98-9 <br><br> PUF (gas phase) | |
| Carbazole/ 86-74-8 <br><br> GFF and PUF | | 3-Iodo-2-propynyl butylcarbamate (Iodocarb, IPBC)/ 55406-53-6 <br><br> PUF (gas phase) | |
| *m*-Terphenyl/ l92-06-8 <br><br> GFF and PUF | | Triallate/ 2303-17-5 <br><br> PUF (gas phase) | |
| *o*-Terphenyl/ 84-15-1 <br><br> PUF (gas phase) | | 2-(Methylmercapto)-benzothiazole (2-Me-S-BTH)/ 615-22-5 <br><br> PUF (gas phase) | |
| *p*-Terphenyl/ l92-94-4 <br><br> PUF (gas phase) | | | |



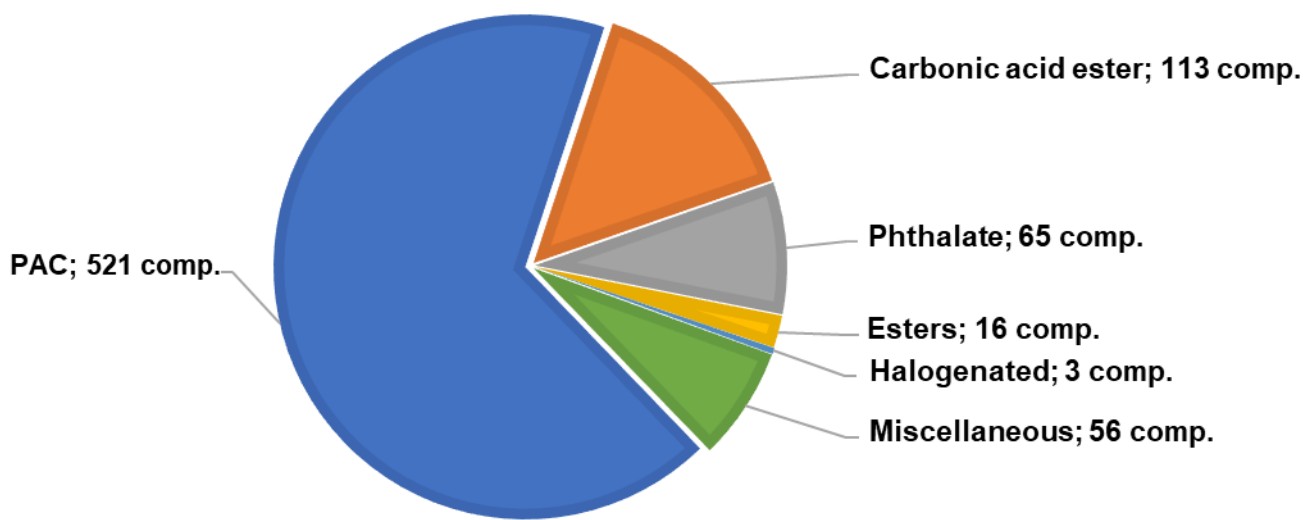

**Figure 6: L3 compounds.**