# Peer review of "Non-target and suspect characterisation of organic contaminants in ambient air, Part I: Combining a novel sample clean-up method with comprehensive two-dimensional gas chromatography"

_Atmospheric Chemistry and Physics, 2020_

## Referee Comment (RC1) · Anonymous Referee #1 · 11 May 2020

The manuscript entitled "Non-target and suspect characterisation of organic contaminants in ambient air, Part I: Combining a novel sample clean-up method with comprehensive two-dimensional gas chromatography" reports a newly clean-up method for the identification of POPs and CECs by GCxGC-MS. The topic is of high interest and the manuscript is well written and organized. Few questions should be addressed before publishing the manuscript: 1. Page 4 lines 18-19. Despite the methodology has been already published, it would be helpful to provide in the manuscript a short description.

[Figure]

2. Page 5 lines 8-9. The quantitative methods must be described, even if they are published, owing to their impact on the results. This information can be given in the supplementary material. 3. Page 7 lines 4-7. Could they be eluted from the column by using a stronger combination solvent? Did the authors try a second elution step? 4. In my opinion, it is not necessary to create a new level of identification. The manuscript levels 0-4 match perfectly with Schymansky's levels 1-5. And the manuscript level 5 used to describe the compound of interest is not needed. Indeed, this scale was developed to describe the level of confidence of the identifications, and "mass spectra of interest" is not an identification level. These compounds can be reported in a separate list. 5. Page 9 lines 1-4. Can the authors explain such a high variability?

---

## Referee Comment (RC2) · Anonymous Referee #2 · 12 May 2020

This manuscript details an analytical methodology to characterise semi-volatile organic (SVOCs) contaminants in air samples. The novelty here is that it uses a bespoke sample clean-up procedure coupled to final-extract analysis using 2-dimensional GC and mass spectrometry for pseudo- (suspect) and non-target chemical identification. The result is a very high level of chemical data acquisition through unique levels of compound resolution resulting in vastly improved identification of SVOCs in air samples. This method lends itself to sensitive, non-target screening of chemicals in environ-

mental samples and the authors use HRMS to assure chemical identification in some cases, which is useful. The method also addresses a major issue of 'which analytical technique' given the growing lists of priority pollutants with widely varying physical-chemical properties, as well as finding 'unknowns'. The authors introduce a hierarchy of chemical grouping based on the confidence of chemical identification, which is a pragmatic approach and is also discussed in some detail with regards to the identification of 'unknowns' for select air samples taken in southern Norway. The concluding statements provide some very useful recommendations regarding the use (or drawbacks) of polyurethane foam as a sampling matrix for SVOCs, and the use of HRMS to qualify/confirm chemical candidates.

I don't see any drawbacks or weaknesses to this manuscript. It's nice to see a comprehensive SI which provides considerable detail to allow other lab groups to reproduce this method. Figure 1 is useful in that it provides a stylised overview to people not familiar with 2-dimensional chromatography, however it would be helpful if the authors could also supply an annotated 2-d chromatogram(s) of their actual air samples (this could be added to the SI).

---

## Author Comment (AC1) · 4 Oct 2020

Kjeller, October 2nd, 2020 ISSUE: Reply to reviewer comments on Manuscript ACP acp-2020-263 entitled "Non-target and suspect characterisation of organic contaminants in ambient air, Part I: Combining a novel sample clean-up method with comprehensive two-dimensional gas chromatography "

Dear Editor, Thank you and the two anonymous reviewers for the constructive and help-

ful comments and suggestions on our manuscript. Please find enclosed our detailed reply on the reviewer comments to our manuscript "Non-target and suspect characterisation of organic contaminants in ambient air, Part I: Combining a novel sample clean-up method with comprehensive two-dimensional gas chromatography ". All suggested changes listed in the reviewer replies are completely addressed. After discussion within the author team, we have listed our final answers and suggestions below.

We thank the reviewers for insightful and constructive comments and hope that our reply is in accordance with their and your expectation.

Sincerely yours Laura Röhler On behalf of the author team

Reply to reviewer comments Anonymous Referee #1 Reviewer comments: 1. Page 4 lines 18-19. Despite the methodology has been already published, it would be helpful to provide in the manuscript a short description.

Author reply The important details for active air sampling are given in the manuscript, further details can be found in the given reference. The used sampling methodology is identical with the standardised sampling equipment which is normally used for routine air monitoring for EMEP, except the here chosen higher sampling rate for the higher air volumes collected.

Reviewer comment 2. Page 5 lines 8-9. The quantitative methods must be described, even if they are published, owing to their impact on the results. This information can be given in the supplementary material.

Author reply We agree with the comment and we have added a short description of these methods in the SI section. We included at page 5, line 9-10: A short description of these methods can also be found in the SI.

Reviewer comment 3. Page 7 lines 4-7. Could they be eluted from the column by using a stronger combination solvent? Did the authors try a second elution step?

Author reply Yes, this is a relevant question. We have, during the method development, tested various solvents, solvent mixtures and adsorbent materials. This included stronger solvents, e.g. hexane/ cyclohexane with ethyl acetate or hexane and toluene. The results showed that they were not applicable, since the tested solvents/solvent mixtures were dissolving and eluting high amounts of the PUF matrix related compounds and creating elevated background signals or high contamination of the analytical systems. These elevated background signals made the comprehensive chromatography-based SUS/NTS impossible. Since SUS/NTS is very labour intensive, the data treatment of several fractions per sample would be unnecessary time consuming. Furthermore, those compounds may even elute and separate into different fractions and, as a consequence, be overlooked in the subsequent data treatment due to low concentrations. We included at page 7, line 8-9: A stronger solvent mixture could not be applied as this results in increased amounts of interfering PUF related matrix compounds in the final extracts (unpublished data).

Reviewer comment 4. In my opinion, it is not necessary to create a new level of identification. The manuscript levels 0-4 match perfectly with Schymansky's levels 1-5. And the manuscript level 5 used to describe the compound of interest is not needed. Indeed, this scale was developed to describe the level of confidence of the identifications, and "mass spectra of interest" is not an identification level. These compounds can be reported in a separate list.

Author reply We agree with the reviewer that this is not needed when working with high resolution MS (HRMS) but since we were applying low-resolution MS (LRMS) GCxGC, to reflect the type of data generated, we needed to slightly modify the identifications criteria for each of the confidence levels (as described in the manuscript). All peaks, which were matching the criteria for SUS and/or NTS during data processing were classified as Level 5, mass spectra of interest, and is an identification level.

Reviewer comment 5. Page 9 lines 1-4. Can the authors explain such a high variability?

Author reply We included at page 6, line 23-26: PUF plugs used for active air sampling

will normally be reused after sample extraction and a complete cleaning procedure. Thus, PUF plugs for sampling and blank samples may be of different age and, thus, the extractable PUF matrix will vary. Extracts from exposed, real high-volume air samples and laboratory blank samples can, thus, contain different peak distribution profiles. Blank filtration strategies are described in sect. 3.2.1.

Anonymous Referee #2 Reviewer comments: Figure 1 is useful in that it provides a stylised overview to people not familiar with 2-dimensional chromatography, however it would be helpful if the authors could also supply an annotated 2-d chromatogram(s) of their actual air samples (this could be added to the SI).

Author reply : We included a 2-D chromatogram to the SI. Included at page 3, line 19 (Figure 1 and Supplementary Information Figure S1)

[Figure]

Figure S1: GC×GC 2-dimensional plot of PUF S2 extract
1: 2-Nitrotoluene; 2: 2-Nitroanisol; 3: Benzensulfonamide; 4: Triallate; 5: Dichlorfluanid

**Fig. 1.**